# Agrochemical Residues in Fish and Bivalves from Sepetiba Bay and Parnaiba River Delta, Brazil

**DOI:** 10.3390/ijerph192315790

**Published:** 2022-11-27

**Authors:** Joyce Aparecida Tavares Miranda, Fabíola Helena S. Fogaça, Sara C. Cunha, Mariana Batha Alonso, João Paulo M. Torres, José Oliveira Fernandes

**Affiliations:** 1Biophysics Institute Carlos Chagas Filho, Universidade Federal do Rio de Janeiro (IBCCF-UFRJ), Av. Carlos Chagas Filho, 373, Rio de Janeiro 21941-902, RJ, Brazil; 2Brazilian Agricultural Research Company, Agroindústria de Alimentos (EMBRAPA), Av. das Américas, nº 29.501, Guaratiba, Rio de Janeiro 23020-470, RJ, Brazil; 3Laboratory of Bromatology and Hydrology (LAQV-REQUIMTE), Faculty of Pharmacy, University of Porto, Rua Jorge de Viterbo Ferreira 228, 4050-313 Porto, Portugal

**Keywords:** organochlorine, organophosphorus, pyrethroids, seafood, MSPD, GC-MS/MS

## Abstract

Accumulation of pesticides has a harmful impact on the environment and human health. The main goal of this work was to develop a method to determine and quantify the residues of thirteen pesticides in edible fish and bivalves such as parati (*Mugil curema*), seabass (*Centropomus* ssp.), mullet (*Mugil brasiliensis*), clams (*Anomalocardia brasiliana*) and mussel (*Mytilus galloprovincialis*) collected from Sepetiba Bay and Parnaiba River Delta (Brazil) between 2019 and 2020. Matrix solid-phase dispersion (MSPD) was used for extraction and quantification through gas chromatography coupled to tandem mass spectrometry (GC-MS/MS). The method was validated (linearity, accuracy and precision) for fatty fish (*Salmo salar*), lean fish (*Mugil curema*) and bivalves (*Mytilus edulis*). The survey found linear correlation coefficients (r) equal to or greater than 0.9 for almost all analytes. The relative standard deviations (RSD) of five replicates were less than 20% for almost all analytes at different concentrations in lean fish, fatty fish and bivalves. Most analytes showed satisfactory accuracy. Alachlor herbicide was found in samples of seabass, mussels, clams and parati with levels ranging between 0.55 to 420.39 μg kg^−1^ dw. Ethion was found in parati (maximum 211.22 μg kg^−1^ dw), mussels (15.1 μg kg^−1^ dw) and clams (maximum 44.50 μg kg^−1^ dw). Alachlor was found in clams (maximum 93.1 μg kg^−1^ dw), and bifenthrin was found in parati (maximum 43.4 μg kg^−1^ dw) and clams (maximum 42.21 μg kg^−1^ dw). The validated method was satisfactory for the determination of eleven pesticides in the fatty fish matrix, and thirteen pesticides in the samples of lean fish and bivalves. The presence of alachlor, ethion and bifenthrin stands out.

## 1. Introduction

Pollution by industrial, domestic and agricultural waste severely affects the aquatic ecosystem. Endocrine disruptors (EDs) are exogenous substances or mixtures that alter functions of the endocrine system and consequently cause adverse health effects in an intact organism, or its progeny, or (sub)populations [1]. ED substances have the ability to alter the functions of the endocrine system and cause adverse effects on human and animal health [2,3,4,5,6,7,8]. Moreover, the EDs may be persistent organic pollutants (POPs). POPs are classes of environmental micropollutants—compounds that are present in the environment in approximate concentrations of µg L^−1^ and ng L^−1^—and the main concerns related to these classes of substances are that they can produce adverse effects in organisms exposed to very low concentrations, remain for decades in the environment, under mechanisms of bioaccumulation and biomagnification, passing through the food chain of the species [3]. The impacts of pesticides on the environment and ecosystems can vary from contamination of groundwater and imbalances in soil microbial populations to the emergence of new pathogens, or imbalance of some existing ones.

Some studies suggest that contamination with chlorinated pesticides can be related to neurodegenerative diseases and cancers and contribute to the development of type 2 diabetes mellitus (T2D), due to the interference in glucose uptake and adiposity function, resulting in proinflammatory responses and disrupted energy metabolism [9,10,11]. These substances can follow the transplacental route, and be transferred through lactation; therefore, individuals can be exposed to these compounds before birth and in early childhood [10,12,13]. Despite the ban on certain organochlorine pesticides (OCPs) in some parts of the world in the early 1970s, these compounds still contaminate the environment because they are POPs [9,10,11,14,15,16]. Exposure to organophosphorus pesticides (OPs) is related to acute neurotoxicity, possibly due to the inhibition of acetylcholinesterase (AChE). Exposure to this agrochemical is also related to attention deficit disorders (ADHD) and autism in children [8,17]. Pyrethroids are highly efficient against a large range of insect pests, with relatively low toxicity for mammalian species [18]. Despite their low human toxicity, there have been several reports of poisoning related to occupational exposure and deliberate ingestion. Pyrethroids act on several channels and proteins, and interfere with the excitatory neurons, inducing cholinergic syndrome (CS), hyperexcitability, choreoathetosis, and profuse salivation; these pesticides are also related to T-syndrome, characterized by tremors [18,19].

Surveys concerning the anthropic impacts caused by POPs, such as pesticides, on aquatic food chains depend on the use of bioindicator species. Bivalves are intensively employed as sentinel bioindicators in various ecological studies, and spatial–temporal assessments of marine and freshwater contaminants [20,21,22,23]. Morphological malformation in the embryos of fish was detected in response to OPs exposure, [24] and there is evidence of tuna hepatic alteration under OCPs exposure [11]. Transplacental transfer of pyrethroid pesticides and OCPs was also observed in aquatic mammals [25,26].

The choice of the analysis method for pesticide residues determination requires consideration of a huge number of factors, such as: chemical class of the analyte and metabolites, and their chemical properties such as boiling point or polarity; appropriate solvent and mobile phases; pesticides’ stability in injector ports, on-column, or in-mass ion sources; chromatographic behavior; interferences in detection; detection method and limit or regulatory requirements. In mass spectrometry methods, one of the most important parameters to consider, particularly when selectivity or sensitivity criteria are required, is the ionization mode. In gas chromatography (GC), the electron ionization (EI) method clearly has an advantage due to the availability of extensive libraries in full scan mode for confirmation of compound identity through library search matching, when sufficient sample concentration is available. GC-MS/MS can provide additional selectivity when compared with the GC/MS method [27]. Regarding extraction techniques, matrix solid-phase dispersion (MSPD) has been successfully used in the extraction of multiple trace pesticide residues analysis in several foods [28]. MSPD is based on thorough mechanical homogenization of the sample with a sorbent, after which the analytes are eluted with an appropriate solvent. MSPD has advantages such as speed, inexpensiveness, simplicity, adaptability, and ease of handling in comparison to traditional techniques such as liquid–liquid extraction [8,21,22,29,30,31,32,33].

As for the environmental concern, aquatic contamination by pesticides can compromise the marine ecosystem. Regarding human nutrition and food security, the exposure of seafood to these compounds in their natural habitats leads to the loss of their quality and compromises their safety for consumption, directly affecting fishing activity. Thus, the objective of the present work was to develop a method for the quantification of 13 pesticide residues belonging to the organochlorine, organophosphorus and pyrethroid classes, in 84 samples of Brazilian seafood from Sepetiba Bay, including parati (*Mugil curema*), seabass (*Centropomus* spp.) and mussel (*Mytilus galloprovincialis*), and from Parnaiba River Delta, including mullet (*Mugil brasiliensis*) and clam (*Anomalocardia brasiliana*) species. Additionally, to demonstrate the presence of pesticides in marine animals for human consumption and the potential deleterious impact on human health and how much it impacts the decisions of control agencies.

## 2. Materials and Methods

### 2.1. Sampling Area

The fish and bivalve samples studied in the present work—seabass, parati, mullet, clams and mussels—were caught by fishermen between 2019 and 2020 in estuaries and adjacent coastal areas of Brazil, a country bathed by the Atlantic Ocean. The collected samples were desiccated, and the edible and nonedible parts were separated. The edible part (muscle) was then stored at approximately −20 °C for 90 days; afterward, the 84 samples were lyophilized and stored in Petri dishes covered with aluminum foil in order to avoid contamination before being transported to Portugal for analysis.

The Parnaiba River Delta is located in the northeast of Brazil, in the state of Piauí. It is considered one of the three largest sedimentary basins in Brazilian territory with an extent of about 2750 km^2^ [34]. The biome is divided between caatinga and tropical forest, housing important plant and animal communities. The tropical climate of the region is characterized by a rainy summer and a dry winter, with 1200 mm annual average rainfall [34,35,36]. The Sepetiba Bay is located on the southeastern coast of Brazil, in the state of Rio de Janeiro, with ~447 km^2^ of extension area at high tide [37]. It is a body of saline and brackish water, which communicates with the Atlantic Ocean through two passages, one on the west and another on the east, as well as having great influence on rivers and tidal channels in its innermost part. The region is highly urbanized with intense industrial activity, such as metallurgical, petrochemical and pyrometallurgical foundries. In this way, the aquatic environment is severely impacted, which in turn contributes to the social impact on the lives of fishing communities, which depend on this activity for subsistence and income [36,38]. Brazil is one of the countries in the world that most uses a wide variety of agrochemicals, so it is expected to find the presence of many classes of these compounds in different Brazilian aquatic biomes.

### 2.2. Reagents and Standards

The compounds analyzed were organophosphorus: lindane, chlorpyrifos-methyl, chlorpyrifos, fipronil, ethion; organochlorines: aldrin, chlordene, hexachlorobenzene, mirex; organochlorine: alachloro; pyrethroids: bifenthrin, permethrin; and the dicarboximide vincozoline. We used a standard compound with purity >98%, supplied by Sigma Aldrich (Saint Louis, MO, USA). Triphenyl phosphate (TPP, purity 99%), purchased from Sigma Aldrich (Saint Louis, MO, USA), was used as an internal standard. 

Acetonitrile and methanol (HPLC grade) were purchased from Honeywell/Riedel-de-Haën TM (Muskegon, MI, USA); acetone, toluene, and carbon tetrachloride from Panreac Quimica SA (Barcelona, Spain). Anhydrous magnesium sulphate and sodium chloride, all reagent grades, were obtained from Honeywell/Fluka (Muskegon, MI, USA), and anhydrous potassium carbonate from Fluka Chemie GmbH (Buchs, France). Ultrapure water (18.2 mΩ cm^–1^) was purified using a Milli-Q gradient system from Millipore (Milford, MA, USA).

Individual standard solutions were prepared at a concentration of 1000 μg mL^–1^ in acetonitrile, acetone, methanol or toluene, according to the analyte solubility. Then, the solutions were stored in amber screw-capped glass flasks in the dark at –20 °C. Working standard solutions at a concentration of 100 μg mL^–1^ were obtained by diluting the stock solutions in acetonitrile and storing them at –20 °C. Mixing solutions of 1.0, 0.5 and 0.25 μg mL^–1^ were prepared in acetonitrile and used to spike the samples in order to obtain matching calibration curves.

### 2.3. Sample Preparation

To extract the analytes, ~500 mg of each sample (dry weight basis, dw) was weighed in amber glass vials (15 mL bottles). One hundred mg of magnesium sulphate, 100 mg of acidified silica [39] and 20 μL of TPP (5 mg L^−1^) were added to the flasks and shaken for 1 min. Then, after mixing overnight (~12 h) at 420 rpm and ~21 °C (D-72379 Hechingen, Edmund Bühler GmbH, Germany), 3.0 mL of acetonitrile: toluene (4:1, *v*/*v*, pH 3.0) was added and mixed for 15 min at 420 rpm at ~21 °C. After centrifugation (3 min at 1500× *g*) 1 mL of supernatant was added to a small aluminum oxide column (100 mg, preactivated with 500 μL of the extracting solvent), and eluted with 500 μL of acetonitrile: toluene (4:1, *v/v*, pH 3.0). Finally, the eluted was evaporated under a gentle stream of nitrogen at 40 °C, then reconstitute to a total volume of 70 μL in toluene, and transferred to a 2 mL amber bottle containing a 100 μL insert. 

### 2.4. Instrumental Analysis and Optimization of the Analytical Method 

The sample extracts (1 µL) were analyzed using a gas chromatograph Agilent 7890B (Agilent Technologies, Santa Clara, CA, USA) coupled to an Agilent 7000C triple quadrupole mass spectrometer (Agilent Technologies, USA), in electron ionization mode (EI), using a 5MS capillary column (30 m × 0.25 mm × 0.25 mm, Agilent J&W USA). The oven was programmed to start at 95 °C, hold for 1.5 min, ramp at 20 °C min^−1^ to 130 °C, ramp again at 5 °C min^−1^ to 230 °C, then ramp at 25 °C min^−1^ to 290 °C, with a total run time of 30 min. The flow rate was 1.3 mL min^−1^ using helium gas as a carrier. The temperatures of the transfer line, ion source, 1st and 2nd quadrupole were 250, 320 and 150 °C, respectively. The collision cell gases were nitrogen (1.5 mL min^−1^) and helium (2.25 mL min^−1^). The triple quadrupole MS was operated in multiple reaction monitoring (MRM) mode detecting two transitions per analyte (Table 1). 

### 2.5. Analytical Method Validation

The defined method was validated by studies of linearity, recovery, repeatability, limit of detection (LOD) and of quantification (LOQ) applied to a fish species with low lipid content (parati, *Mugil curema*), a species of fatty fish (*Salmon salar*), and a species of bivalve (mussels). Linearity was evaluated by calibration-matched curves. For this purpose, an analytical curve was obtained with spiked matrices (free of analytes), with levels ranging from 10 to 840 μg kg^−1^ dw of the analytes (between 5 and 8 different concentrations depending to the analyte). The limit of detection (LOD) and limit of quantification (LOQ) were calculated, by adopting the signal/noise ratio of 3:1 and 10:1, respectively, using the relative standard deviation (RSD) of ten injections from the low calibration level. Microsoft Excel 2016™ software was applied in the validation methods. 

### 2.6. Determination of Analytes in Samples

Samples of seabass (*Centropomus* spp., *n* = 16), parati (*Mugil curema*, *n* = 17), and blue mussel (*Mytilus edulis*, *n* = 6 pools of 12 units each) from Sepetiba Bay; and mullet (*Mugil* spp., *n* = 39) and clams (*Anomalocardia brasiliana*, *n* = 19 pools of 60 units each) from Parnaiba River Delta were analyzed by GC-MS/MS, according to the description in Section 2.3 and Section 2.4. The differences between the concentrations of compounds present in the samples were compared using Tukey’s *t*-test, with 5% error probability, using Graphpad Prism™ software.

## 3. Results and Discussion

### 3.1. Optimization of Extraction Method

Linearity, represented by linear correlation coefficients (r), was equal to or greater than 0.99 for almost all pesticides found in the three different matrices, except for fipronil in fatty fish (salmon). However, in lean fish species, vinclozolin, alachlor, chlorpyrifos, mirex and bifenthrin had coefficients lower than 0.9. Concerning the precision interlay for the lean fish matrix, the relative standard deviations (RSD) of five replicates were less than 20% for almost all analytes at different levels, except fipronil at 320 μg kg^−1^ dw, showing satisfactory repeatability of the method. In fatty fish, RSD was higher than 20% for all pesticides in at least some of the dilutions analyzed, and for lindane and mirex in all dilutions. In bivalves, RSD was higher than 20% for chlorpyrifos-methyl, chlorpyrifos, alachlor, mirex and atrazine in at least some of the dilutions analyzed (Table 2). 

The method verified using mackerel fillet by Wongmaneepratip, Leong and Yanga [40] for pyrethroids achieved standard deviations (SD) of 4.91 to 13.69%. The majority of agrochemicals analyzed by Petrarca et al. [41] and Kaczyński et al. [42] achieved a repeatability RSD ≤ 20% in the analysis of agrochemicals in seafood.

Regarding accuracy studies, by standard recovery, at a concentration of 10 μg kg^−1^ for lean fish, most analytes showed an extractive yield (%) greater or less than the accepted margin (70 to 120%) [43], indicating that in this concentration the analytes do not present reliable values. For this reason, the recovery of the other matrices was analyzed from 20 μg kg^−1^ dw. In the other concentrations, the vast majority of the compounds that could be analyzed showed a yield between 70 and 120%, except chlorpyrifos-methyl at 80 μg kg^−1^ dw, and atrazine at 320 μg kg^−1^ dw, as shown in Table 3.

Concerning the fatty fish matrix, most analytes showed extractive yields (%) within the accepted range (70 to 120%) [43], except for ethion, mirex and bifenthrin at 270 μg kg^−1^ and permethrin at all analyzed concentrations. Table 4 presents these values.

Regarding the bivalve samples, many pesticides showed an extractive yield (%) above the accepted margin by 20 μg kg^−1^ dw [43]. At 120 μg kg^−1^ dw the findings were satisfactory, except for atrazine. At 400 μg kg^−1^ dw only aldrin, mirex and permethrin had unsatisfactory yields (Table 5).

The mean concentration of organochlorine recoveries in spiked fish samples was 91% with RSD < 20% in the Cho, Lee and Jung [44] study. The majority of agrochemicals presented mean recoveries within the range of 70–120% and repeatability RSD ≤ 20% in bivalves, lean and fatty fishes [41]. The method proposed by Wongmaneepratip, Leong and Yanga [40] applied to pyrethroid analysis in salmon, seabass, threadfin fish, tiger prawn, vannamei prawn, shrimp, squid, grand jackknife clam, and oyster provided recoveries of 75.95–96.81%. Table 6 presents LOD and LOQ values for the three matrices.

In the trace analysis of organic pollutants in seafood, it is very important to remove some compounds, such as lipids, in order to decrease interference with the analysis, to achieve a low detection limit and to protect the analytical instruments [44]. In the present work, the values of LOD and LOQ found in each analyte of lean fish were between 0.4 μg kg^−1^ (HCB) and 21.2 μg kg^−1^ (ethion) and from 1.3 μg kg^−1^ (HCB) to 64.4 μg kg^−1^ (ethion), respectively. The values found for LOD and LOQ in each analyte of fatty fish were from 8.2 μg kg^−1^ (lindane) to 20.3 μg kg^−1^ (fipronil) and 24.9 μg kg^−1^ (lindane) to 61.4 μg kg^−1^ (fipronil), respectively. In the bivalve matrix, the values of LOD and LOQ found in each analyte varied from 0.1 μg kg^−1^ to 0.4 μg kg^−1^ (lindane and mirex), respectively. However, for permethrin in the bivalve matrix, the values found for LOD and LOQ were 280.1 μg kg^−1^ and 484.7 μg kg^−1^, respectively, suggesting that the extraction method was not effective for this compound determination and quantification from bivalve matrices.

Kaczyński et al. [42] obtained LOD values ranging from 0.05 to 1.2 μg kg^−1^ for fishes’ muscles and liver. Wongmaneepratip, Leong and Yanga [40] achieved LOQs of 0.54–0.85 ng g^−1^ for pyrethroids in seafood. The method’s detection limits ranged from 0.05 to 1.2 μg kg^−1^ in the Petrarca et al. [41] study.

### 3.2. Determination of Analytes in Samples

The presence of alachlor, ethion, bifenthrin and permethrin was determined by analyzing some samples. The herbicide alachlor was found in four seabass samples (levels from 6.77 to 32.36 μg kg^−1^ dw), in five mussel samples (levels from 0.66 to 33.41 μg kg^−1^ dw), eighteen clam samples (levels from 0.55 and 93.17 μg kg^−1^ dw), and in three parati samples (levels from 8.90, 9.01 and 420.39 μg kg^−1^ dw). The parati samples that were more contaminated with alachlor was also found to contain ethion with a level of 211.22 μg kg^−1^ dw. The presence of ethion was also observed in three samples of mussels, with levels ranging from LOQ to 15.13 μg kg^−1^ dw, and in 13 samples of clams, with levels ranging from LOQ to 44.50 μg kg^−1^ dw. Regarding the pyrethroid class, permethrin and bifenthrin were the pesticide residues analyzed. Permethrin was observed in two samples of mullet with levels below the LOQ. Bifenthrin was found in four samples of parati, two of them with levels above the LOQ, in concentrations of 41.01 and 43.41 μg kg^−1^ dw. Bifenthrin was also found in 23 samples of mullet, but only two with levels above the LOQ, in concentrations of 32.14 and 36.06 μg kg^−1^ dw, and in two samples of clams, only one of which had a value above the LOQ, in a concentration of 42.20 μg kg^−1^ dw. There are no differences between the concentrations of compounds present in the samples, compared using Tukey’s *t*-test, with 5% error probability. Table 7 shows the levels of the analytes found in the samples (in μg kg^−1^ dw).

The European Union (EU) recommends an MLR of 0.01 mg kg^−1^ for tall oil repellents in fish, fish products and any other marine and freshwater food products [43], which is the case of bifenthrin in parati and clams. There is literature on the lethal and sublethal effects of these agrochemical classes on fish, including alteration of behavior (sensation, locomotion, feeding, learning, etc.), physiology (respiration, metabolism, reproduction, etc.), alteration of biochemistry of enzymes, blood and hormones, carcinogenicity and mutagenicity [45,46].

A mutagenic response has been observed for alachlor metabolites as ethanesulfonic and oxanilic acid, which are more persistent in soil than alachlor itself [47]. The ethion metabolite monoxon may be hazardous to human health [48]. Regarding the pyrethroid bifenthrin, its metabolites, such as benzene 1,1(methylthio)ethylidene, may be less toxic than bifenthrin for some organisms [49].

Pesticides have caused serious health problems as they accumulate in fat-rich foods such as meat and milk and affect the food chain. Human exposure to these pesticides could occur through diet, including food items such as meat, fish, poultry, and dairy products [50]. Fishermen and coastal populations are especially exposed to the chronic effects of dietary pesticide ingestion, such as hypertension, cardiovascular disorders, endocrine system disorders and other health-related problems in humans.

The basic characteristics of organochlorine pesticides are high persistence, low polarity, low aqueous solubility, and high lipid solubility. Their toxicity is mainly due to stimulation of the central nervous system, as cyclodines, such as the GABA antagonists endosulphan and lindane, inhibit the calcium ion influx and Ca- and Mg-ATPase causing the release of neurotransmittors. Epidemiological studies have exposed the etiological relationship between Parkinson’s disease and organochlorine pollutants [51].

Pyrethroid insecticides are also reported to be found in seafood. Rocha et al. [52] identified transfluthrin, cyhalothrin, permethrin, cyfluthrin, cypermethrin, fenvalerate, bifenthrin, phenothrin, fluvalinate, deltamethrin, and flumethrin in the roe or gonads of sea urchins (*Paracentrotus lividus*) from the northwest (NW) Portuguese coast, and Gadelha et al. [53] reported five pyrethroid (tetramethrin, bifenthrin, cyhalothrin, fenvalerate and permethrin) insecticides in farmed oysters from Aveiro (Portugal, north coast). Riaz et al. [54] determined the concentrations of cypermethrin, deltamethrin, permethrin, and bifenthrin in the samples of water, sediments, and fish collected from various locations of the Chenab River, Pakistan, during summer and winter seasons. In their study, the highest mean concentrations of pyrethroid (1248 μg g^−1^ dw) were detected in fish collected in winter, as compared with summer (0.087 μg L^−1^ dw).

Traces of organophosphorus pesticides (OPs) present in fish fed in aquaculture can affect the organisms [24,55,56]. Fenthion (0.0026–0.0037 mg kg^−1^) exhibited values above the limit of quantification in tilapia from Aracoiaba, Castanhão, and Ilha Solteira [57]. Martins, Costa & Bianchini [58] found a maximum concentration 4.93 ng g^−1^ of chlorpyriphos in the muscle of Brazilian guitarfish (*Pseudobatos horkelli*). Zhang et al. [13] found Fipronil sulfone in 16 seafood samples (ten shrimp and six fish) with concentrations ranging from 17.2 μg kg^−1^ to 181.5 μg kg^−1^, and in two shrimp samples, above 50.3 μg kg^−1^.

Several authors around the world reported the presence of pesticides in seafood [8]. The OCPs class of pesticides, especially DDE, a DDT metabolite, is commonly found in muscle samples of fish such as tuna (*Thunnus* spp.), swordfish (*Xiphias gladius*), king mackerel (*Scomberomorus cavalla*), sardines (*Sardinella* spp.), corvina (*Argyrosomus regius*), mullet, mackerel (*Trachurus* spp.), mussels, blue crabs (*Callinectes sapidus*), clams, tilapia (*Oreochromis niloticus*), bigeye barracuda (*Sphyraena forsteri*), and cod fish (*Gadus morhua*). Despite the fact that some OCP compounds have been restricted and banned, isomers of DDT are frequently detected in top predators, due to bioaccumulation and biomagnification along with the food web [59,60,61,62,63,64,65,66,67,68,69].

Moraes [70] described a scientific trend of studies on the effect of pesticides on the fish endocrine system. Data were compiled through the Web of Science, Scopus and Scielo databases, through a diagnosis that identified studies from 1970 to 2017. The author identified a significant increase in publications for this period (r = 0.83; *p* < 0.0001). It was observed that most of the studies took place ex situ, and pesticides such as endosulfan, dichlorodiphenyltrichloroethane (DDT), together with its metabolic dichlorodiphenyldichloroethylene (DDE), atrazine, and chlorpyrifos were the most investigated. On the other hand, the groups of contaminants most studied in situ were organochlorines, organophosphates and alkylphenols. Countries like the United States, China and India dominated the polls. The most used species in the research were zebrafish (*Danio rerio*), rainbow trout (*Oncorhynchus mykiss*) and medaka (*Oryzias latipes*). Effects such as hormonal and histological changes, genetic expressions and gonadal damage were significant in fish in the presence of these chemicals. According to the author, this area is still a little lacking in research for endocrine analysis.

## 4. Conclusions

Considering that the indiscriminate use of pesticides can contaminate groundwater, and that the inappropriate discharge of effluents results in rivers, seas and oceans being the final destinations of these residues, studies on the contamination of aquatic biota are necessary, especially concerning biomagnification among species. Through the application of this validated method, it was possible to determine 11 pesticides in the fatty fish matrix, and 13 in the lean fish and bivalve matrices by GC-MS/MS. Hence, in this study the presence of three concerning classes of pesticides, organochlorine, organophosphorus and pyrethroids, was found in seafood samples. The solid-phase matrix dispersion method for analyte extraction, associated with mass spectrometry for detection and quantification of agrochemicals, provided a 95% reduction in analysis time and in the amount of solvent used compared with conventional methods. In addition, quick and multianalyte techniques for the detection and prevention of chemical and biological contaminants in fresh and industrialized foods can contribute to government programs and the productive sector, in the control and mitigation of food contamination, strengthening safety and zoosanitary defense.

## Figures and Tables

**Table 1 ijerph-19-15790-t001:** GC-MS/MS parameters for determination of target analytes by MRM.

Target Compounds	Log K_ow_	Transition *m/z*		Collision Energy (KV)
Lindane	3.72	218.8> 181> 180.9>	183 109 145	5 30 12
Chlorpyriphos-methyl	4.31	288> 286>	93 271 270.0	15 26 16
Chlorpyriphos	4.96	318.8> 314> 313.8> 28.8>	286 258 272.9	5 14 15
Fipronil	4.00	367>	228 225 224 213	30 25 20 30
Ethion	5.07	231>	175 129	5 25
Aldrin	6.50	298> 263> 257>	263 191 193 222	8 30 12
Chlordene	5.57	230>	195 160	25 40
HCB	5.73	283.9> 284>	248.8 213.9 142	25 35 50
Mirex	6.89	272> 271.9>	237 167 235 116.9	20 40 15 25
Bifenthrin	6.00	166> 181> 182>	165 166 167	16 25 12
Permethrin	6.50	183.1> 183> 163>	168.1 153.1 115.2 77.1 127	15 25 38 5
Alachlor	3.52	270> 238>	161.5	20
Vinclozolin	3.10	187	145.0 172.2	15

**Table 2 ijerph-19-15790-t002:** RSD, referring to repeatability of the analysis method in three concentrations for the lean fish, fatty fish and bivalve matrices.

	Lean Fish (μg kg^−1^ dw)			Fatty Fish (μg kg^−1^ dw)			Mussel (μg kg^−1^ dw)		
	10	80	320	20	120	270	20	120	400
Lindane	1.7	1.0	5.4	43.8	22.3	27.3	6.5	12.8	4.3
Chlorpyrifos-methyl	0.8	2.3	3.2	15.8	40.6	32.3	26.6	12.6	7.5
Chlorpyrifos	0.4	1.5	5.8	40.0	16.0	20.4	33.0	13.4	2.4
Fipronil	5.8	5.4	41.0	33.5	8.8	18.7	20.12	7.4	6.0
Ethion	0.7	2.6	15.4	36.9	20.5	14.5	18.16	80.0	7.5
Aldrin	1.2	2.2	5.7	29.4	14.6	14.9	16.9	10.9	9.3
Chlordene	3.4	2.5	4.0	25.5	14.3	23.8	9.4	10.4	9.0
HCB	1.9	2.5	4.2	41.3	7.2	14.3	16.6	15.8	3.6
Mirex	2.6	1.2	3.0	22.7	24.2	30.4	23.4	26.4	44.7
Bifenthrin	1.1	1.3	2.4	21.6	26.4	19.1	8.4	6.8	3.6
Permethrin	10.1	3.1	2.2	16.8	26.9	32.9	11.1	20.5	30.6
Alachlor	0.9	2.3	2.1	-	-	-	25.0	105.3	31.9
Vinclozolin	3.3	3.8	4.0	7.2	9.3	20.1	17.8	16.0	6.6

**Table 3 ijerph-19-15790-t003:** Interval (minimum–maximum) of mean extractive recoveries (%) and RSD for the analytes at three concentrations for the lean fish matrix.

	10 μg kg^−1^ dw		80 μg kg^−1^ dw		320 μg kg^−1^ dw	
	Recovery (%)	RSD (%)	Recovery (%)	RSD (%)	Recovery (%)	RSD (%)
Lindane	81.0	1.2	81.9	2.6	102.5	5.5
Chlorpyrifos-me	85.7	0.2	44.3	58.1	101.7	1.9
Chlorpyrifos	55.3	8.0	89.6	2.5	103.1	1.6
Fipronil	45.3	18.4	77.9	14.2	81.8	18.9
Ethion	103.4	4.9	84.9	7.2	94.5	4.9
Aldrin	46.3	2.2	96.9	3.8	104.6	10.5
Chlordene	50.3	14.5	100.8	4.6	99.4	1.5
HCB	98.1	4.8	92.1	4.0	99.9	4.6
Mirex	7.3	7.8	101.9	0.8	103.9	1.3
Bifenthrin	35.1	10.3	98.3	3.3	99.1	1.4
Permethrin	89.6	12.2	120.8	21.9	113.6	9.7
Alachlor	59.3	0.1	88.9	1.8	98.6	1.5
Vinclozolin	44.4	10.1	102.0	3.5	111.9	4.4

**Table 4 ijerph-19-15790-t004:** Interval (minimum–maximum) of mean extractive recoveries (%) and RSD for the analytes at three concentrations for the fatty fish matrix.

	20 μg kg^−1^ dw		120 μg kg^−1^ dw		270 μg kg^−1^ dw	
	Recovery (%)	RSD (%)	Recovery (%)	RSD (%)	Recovery (%)	RSD (%)
Lindane	69.6	12.3	84.9	10.5	79.6	12.0
Chlorpyrifos-me	88.5	14.6	74.2	10.2	71.7	25.0
Chlorpyrifos	97.2	7.3	109.1	11.1	86.3	24.4
Fipronil	105.3	0.9	87.3	3.7	92.4	8.9
Ethion	85.5	2.3	79.4	38.2	68.5	64.2
Aldrin	89.3	2.4	87.9	1.9	82.7	39.8
Chlordene	74.7	1.3	90.5	37.2	71.3	48.9
HCB	85.4	11.2	103.5	10.4	89.3	5.6
Mirex	65.9	0.3	71.5	64.4	57.8	79.3
Bifenthrin	79.8	1.0	88.1	47.1	62.8	66.2
Permethrin	64.7	0.6	55.1	70.4	39.4	121.2
Vinclozolin	95.7	0.4	82.5	6.6	75.6	21.7

**Table 5 ijerph-19-15790-t005:** Interval (minimum–maximum) of mean extractive recoveries (%) and RSD for the analytes at three concentrations for the bivalve matrix.

	20 μg kg^−1^ dw		120 μg kg^−1^ dw		400 μg kg^−1^ dw	
	Recovery (%)	RSD (%)	Recovery (%)	RSD (%)	Recovery (%)	RSD (%)
Lindane	106.3	2.7	97.0	4.3	99.5	5.5
Chlorpyrifos-me	115.5	12.5	105.9	7.5	99.0	14.9
Chlorpyrifos	114.2	5.4	100.4	2.3	102.8	20.0
Fipronil	107.4	3.2	74.5	6.0	77.1	28.8
Ethion	91.8	9.1	88.5	7.5	96.6	4.2
Aldrin	102.7	10.0	111.5	9.3	10.0	7.0
Chlordene	108.7	18.3	64.8	9.0	86.3	14.7
HCB	170.3	9.2	93.0	3.6	97.2	10.7
Mirex	105.7	40.3	63.4	44.8	51.3	92.2
Bifenthrin	99.1	5.7	98.0	3.6	101.4	10.6
Permethrin	370.8	35.6	106.1	30.6	50.5	165.7
Alachlor	99.3	2.6	95.4	31.9	70.6	3.9
Vinclozolin	131.3	34.8	83.3	6.6	109.9	10.5

**Table 6 ijerph-19-15790-t006:** LOD and LOQ for the three matrices.

	Lean Fish (μg kg^−1^ dw)		Fatty Fish (μg kg^−1^ dw)		Mussel (μg kg^−1^ dw)	
	LOD	LOQ	LOD	LOQ	LOD	LOQ
Lindane	1.5	4.5	8.2	24.9	0.1	0.4
Chlorpyrifos-me	0.5	1.5	12.5	37.9	7.0	21.2
Chlorpyrifos	4.3	13.2	10.4	31.4	6.9	20.8
Fipronil	1.9	5.7	20.3	61.4	15.9	48.3
Ethion	21.2	64.4	12.8	38.5	1.5	4.6
Aldrin	6.3	18.9	13.3	40.2	6.0	18.3
Chlordene	9.5	28.7	10.4	31.5	9.0	27.2
HCB	0.4	1.3	10.6	32.0	21.0	63.2
Mirex	12.1	36.6	11.9	36.1	0.1	0.4
Bifenthrin	12.3	37.2	12.9	30.1	37.2	39.1
Permethrin	17.2	52.3	11.54	35.0	280.1	484.7
Alachlor	5.2	15.7	-	-	0.5	1.4
Vinclozolin	8.4	25.3	16.3	49.4	0.5	1.4

**Table 7 ijerph-19-15790-t007:** Concentrations of analytes found in the analyzed samples (μg kg^−1^ dw).

Found Analyte	Class	Seafood Species	Concentrations
Alachlor	Herbicide	seabass (*n* = 4)	6.8 to 32.4 μg kg^−1^
		mussels (*n* = 5)	0.7 to 33.4 μg kg^−1^
		clams (*n* = 18)	0.5 to 93.1 μg kg^−1^
		parati (*n* = 3)	8.9; 9.0 and 420.4 μg kg^−1^
Ethion	organophosphorus	parati (*n* = 1)	211.2 μg kg^−1^
		mussels (*n* = 3)	<LOQ (4.6 μg kg^−1^), 12.9 and 15.1 μg kg^−1^
		clams (*n* = 13)	<LOQ to 44.5 μg kg^−1^
Permethrin	Pyrethroid	clams (*n* = 2)	<LOQ
Bifenthrin		parati (*n* = 4)	<LOQ to 43.4 μg kg^−1^
		mullet (*n* = 23)	<LOQ
		clams (*n* = 2)	<LOQ and 42.2 μg kg^−1^

## Data Availability

Not applicable.

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
