# Peer review of "Agrochemical Residues in Fish and Bivalves from Sepetiba Bay and Parnaiba River Delta, Brazil"

_ijerph, 2022, doi:10.3390/ijerph192315790_

Round 1

Reviewer 1 Report

The manuscript is well written and aims to develop a method to perform the determination and quantification of 13 pesticide residues in Brazilian seafood samples.
The materials and techniques are suitable for the proposed objective. Analytical precautions are well described for quality assurance
The presentation of the data is very detailed and the discussion well argued. Just a suggestion. Although the focus is on the development and optimization of methods, it would be useful to compare the results with the safety values of pesticides in fish and/or seafood proposed by regulatory agencies in Brazil or abroad, such as the Food and Drug Administration (FDA), for example.
I consider the manuscript a good contribution to the study of environmental contaminants and food safety.

Author Response

Dear Reviewer,

Thank you for the review, your suggestion can be found right after table 7.

Best wishes

Reviewer 2 Report

Int. J. Environ. Res. Public Health

Article: ijerph-1998607

Agrochemicals Residues in Fishes and Bivalves From Sepetiba 2 Bay and Parnaiba River Delta, Brazil

The article describes the identification by gas chromatography of pesticides in fish and bivalves such as parati (Mugil curema), seabass (Centropomus ssp.), mullet (Mugil brasiliensis), clams (Anomalocardia brasiliana) and mussel (Mytilus galloprovin-cialis) collected from Sepetiba Bay and Parnaiba River Delta (Brazil) between 2019 and 2020.

Specific comments:

1. For a better understanding, it is necessary to make some changes in the text architecture in order to clarify the purpose of the manuscript:

·         Demonstrate the importance of extraction methods (Matrix solid-phase dispersion - MSPD) and quantification (gas chromatography coupled to tandem mass spectrometry -GC-MS/MS) for the identification of different types of pesticides in animal samples?

·         Or validate the analytical method using the fish and bivalves samples?

·         Or demonstrate the presence of pesticides in marine animals for human consumption and the potential deleterious impact on human health and how much it impacts the decisions of control agencies.

2. Make it clear whether there is a direct relationship between the choice of these pesticides analyzed and the regions used for sample collection. Include data showing the significant presence of these pesticides in these regions (Sepetiba Bay and Parnaiba River Delta, Brazil).

3. Emphasize the importance of the amounts of pesticides found in different species in terms of their ability to induce toxicity in marine animals and intoxication in human consumers, taking into account their intake in the total diet and whether the potential to accumulate.

4. Mention if the levels of pesticides found will undergo some type of metabolic modification in the species in order to produce more or less toxic metabolites.

Author Response

Follow attached...

Reviewer 3 Report

The study entitled Agrochemicals Residues in Fishes and Bivalves From Sepetiba 2 Bay and Parnaiba River Delta, Brazil, by Miranda et al. aimed to develop a method to determine and quantify thirteen pesticides 11 residues in edible fishes and bivalves. This work is interesting and important. The methods align with the study aim and the design is good. The literature review is up-to-date. However, the organization could be better and some parts could be clearer. 

General comment

There is a general need for formatting, there are extra spaces in various parts of the text, in addition, the units of measurement are sometimes with uppercase K, sometimes with lowercase K. In this case, the K must be lowercase.

Also, different letter fonts were used, please standardize.

Abstract

Please put the species' name in italics.

Lines 23-24 rewrite this sentence more clearly, please.

Superscript the -1 of the unit of measurement, please.

Material and method

In the sampling area, please provide more information about the year of sampling and the sample number

What edible parts were selected? Muscle? Please explain more clearly.

Material and methods lacked an explanation of the statistical tests, and which software was used.

Results and discussion

Section 3.1 went straight to 3.3, thus missing section 3.2.

Conclusion

I found the conclusion too simplistic, it should be improved. It should be clarified for which equipment this analysis was validated. And answer the following question: How does this research compare to what was already known from other studies on this topic?

Author Response

follow attached...

Round 2

Reviewer 2 Report

The authors made some changes to the text in response to the suggested remarks. With a few more fine adjustments to the spellings, the text will be ready for publication.

Reviewer 3 Report

I recommend accepting the paper, the changes made by the authors were satisfactory. Thank you!